# The Use of Confocal Laser Endomicroscopy in Diagnosing Barrett’s Esophagus and Esophageal Adenocarcinoma

**DOI:** 10.3390/diagnostics12071616

**Published:** 2022-07-02

**Authors:** Jitka Vaculová, Radek Kroupa, Zdeněk Kala, Jiří Dolina, Tomáš Grolich, Jakub Vlažný, David Said, Lydie Izakovičová Hollá, Petra Bořilová Linhartová, Vladimír Procházka, Marek Joukal, Petr Jabandžiev, Ondřej Slabý, Lumír Kunovský

**Affiliations:** 1Department of Gastroenterology and Internal Medicine, University Hospital Brno, Faculty of Medicine, Masaryk University, 62500 Brno, Czech Republic; vaculova.jitka@fnbrno.cz (J.V.); kroupa.radek@fnbrno.cz (R.K.); dolina.jiri@fnbrno.cz (J.D.); 2Department of Surgery, University Hospital Brno, Faculty of Medicine, Masaryk University, 62500 Brno, Czech Republic; kala.zdenek@fnbrno.cz (Z.K.); grolich.tomas@fnbrno.cz (T.G.); prochazka.vladimir@fnbrno.cz (V.P.); 3Department of Pathology, University Hospital Brno, Faculty of Medicine, Masaryk University, 62500 Brno, Czech Republic; vlazny.jakub@fnbrno.cz (J.V.); said.david@fnbrno.cz (D.S.); 4Department of Pathophysiology, Faculty of Medicine, Masaryk University, 62500 Brno, Czech Republic; lydie.izakovicova-holla@fnusa.cz (L.I.H.); peta.linhartova@gmail.com (P.B.L.); 5Clinic of Stomatology, St. Anne’s University Hospital, Faculty of Medicine, Masaryk University, 65691 Brno, Czech Republic; 6RECETOX, Faculty of Science, Masaryk University, Kotlarska 2, 60200 Brno, Czech Republic; 7Clinic of Maxillofacial Surgery, University Hospital Brno, Faculty of Medicine, Masaryk University, Jihlavska 20, 62500 Brno, Czech Republic; 8Department of Anatomy, Faculty of Medicine, Masaryk University, 62500 Brno, Czech Republic; mjoukal@med.muni.cz; 9Department of Pediatrics, University Hospital Brno, Faculty of Medicine, Masaryk University, 61300 Brno, Czech Republic; jabandziev.petr@fnbrno.cz; 10Central European Institute of Technology, Masaryk University, 62500 Brno, Czech Republic; on.slaby@gmail.com; 11Department of Biology, Faculty of Medicine, Masaryk University, 62500 Brno, Czech Republic; 122^nd^ Department of Internal Medicine–Gastroenterology and Geriatrics, University Hospital Olomouc and Faculty of Medicine, Palacky University Olomouc, 77900 Olomouc, Czech Republic; 13Department of Gastroenterology and Digestive Endoscopy, Masaryk Memorial Cancer Institute, 65653 Brno, Czech Republic

**Keywords:** confocal laser endomicroscopy, probe-based confocal laser endomicroscopy, diagnosis, Barrett’s esophagus, esophageal cancer, early esophageal adenocarcinoma

## Abstract

Confocal laser endomicroscopy (CLE) is a diagnostic technique that enables real-time microscopic imaging during microscopic examination and evaluation of epithelial structures with 1000-fold magnification. CLE can be used in the diagnosis of various pathologies, in pneumology, and in urology, and it is very widely utilized in gastroenterology, most importantly in the diagnosis of Barrett’s esophagus (BE), esophageal adenocarcinoma (EAC), biliary strictures, and cystic pancreatic lesions. A literature search was made in MEDLINE/PubMed and Google Scholar databases while focusing on diagnostics using CLE of BE and EAC. We then examined randomized and observational studies, systematic reviews, and meta-analyses relating to the utilization of CLE in BE and EAC diagnostics. Here, we discuss whether CLE can be a suitable diagnostic method for surveillance of BE. Even though many studies have proven that CLE increases diagnostic accuracy in detecting neoplastic transformation of BE, CLE is still not used as a standard diagnostic tool in BE surveillance due to a deficiency of scientific evidence. More studies and data are needed if CLE is to find a place as a new technique in BE surveillance.

## 1. Introduction

Confocal laser endomicroscopy (CLE) is an endoscopic-assisted technique that enables the obtaining of very high magnification of the mucosal layer of the gastrointestinal tract, allows imaging of cellular and subcellular details, and can also provide histological diagnosis in real-time. There are two types of CLE: endoscope-based confocal laser endomicroscopy (eCLE) and probe-based confocal laser endomicroscopy (pCLE). To perform eCLE, a dedicated endoscope with a miniaturized confocal scanner integrated into the distal tip is employed [1,2,3]. Several studies have been published using eCLE, but this system is no longer commercially available. pCLE uses a separate unit, which is placed outside the endoscope and emits the laser for the imaging. During endoscopy the special probe is inserted into the scope channel [4]. pCLE is gaining in popularity for gastroenterology, and it can be used in the diagnosis of Barrett’s esophagus (BE), inflammatory bowel disease, biliary strictures, pancreatic cystic lesions, and colorectal lesions [5,6]. Moreover, it can be useful in the diagnosis of celiac disease, microscopic colitis, and *Helicobacter pylori* chronic gastritis [7]. pCLE can also be used intraoperatively. In 2019, Fuks et al. published a study that enrolled 21 patients. All were diagnosed with digestive cancers and underwent surgical resection or an exploratory laparoscopy during which microscopic images were acquired from different tissues (peritoneum, liver, lymph node, diaphragm, colon, and adrenal gland). Microscopic images were obtained using a combination of pCLE with a specifically designed device with a bending distal tip providing easy access to abdominal organs. According to the study, real-time intraoperative pCLE is a feasible and safe method that could provide valuable information intraoperatively [8]. Several studies have also shown the use of pCLE in diagnosis of brain tumors, melanoma, and oral cavity cancer [9]. 

A consensus approach among pCLE users for standardization of image criteria is termed the “Miami classification”. It encompasses consensus for the likes of BE, biliary diseases, colorectal diseases (colorectal polyps and ulcerative colitis), as well as gastric and duodenal diseases [10,11]. The most commonly used classifications for evaluating BE’s dysplasia, however, are from di Pietro et al. [12] for low-grade dysplasia (LGD) and Gaddam et al. [13] for high-grade dysplasia (HGD)/esophageal adenocarcinoma (EAC). Diagnostic criteria from di Pietro et al. [12] were published in 2019. The best cutoff for LGD diagnosis is the positivity of any three of the following six criteria: dark non-round glands, irregular gland shape, lack of cells goblet, sharp cutoff of darkness, variable cell size, and cellular stratification. In 2011, Gaddam et al. [13] established the pCLE criteria for dysplastic BE (HGD/EAC). Their study resulted in the formulation of a total of six pCLE criteria that predicted dysplasia with a good degree of accuracy. These criteria were as follows: saw-toothed epithelial surface, not-easily identifiable goblet cells, non-equidistant glands, unequal size and shape of glands, enlarged cells, and irregular and non-equidistant cells. The criteria for each diagnosis are presented in Table 1.

In this review, we focus on pCLE in the diagnosis of BE and EAC. BE is the most-known risk factor for the development of EAC, and the risk of EAC is greater by about 30-fold or more among patients with BE compared to that for the general population [14,15]. BE occurs when the normal squamous epithelium of the distal esophagus (Figure 1a–c) changes to columnar-lined intestinal-type cells (Figure 2a,b), which can usually be seen when the mucosa of the esophagus is repeatedly exposed to gastric acid [16,17,18]. Other risk factors for the development of BE are male gender, age, race, smoking, alcohol consumption, and obesity [19,20]. BE is a premalignant lesion that develops in 6–14% of patients with gastroesophageal reflux disease (GERD), approximately 0.5–1% of whom will develop adenocarcinoma [21]. The risk factors for EAC are very similar to those for BE, mainly GERD, cigarette smoking, and obesity. The risk of EAC also increases with age and is more likely in the male gender and Caucasians [22,23,24,25].

The main diagnostic method for BE is esophagogastroduodenoscopy (Figure 3a,b) with esophageal biopsies for histological examination. Finding of salmon-colored mucosa in the esophagus is typical for BE (segment ≥ 1 cm), while intestinal metaplasia is confirmed histopathologically [26,27]. Dysplastic BE includes LGD and HGD with higher risk of progression to EAC (Figure 4a,b). According to the latest European Society for Gastrointestinal Endoscopy (ESGE) guidelines, high-definition white-light endoscopy (HD-WLE) is highly recommended for use in BE surveillance. Routine use of advanced endoscopy imaging, including CLE, is not yet recommended [28]. The tissue sampling should be according to the Seattle Protocol (four-quadrant biopsies at 1–2 cm intervals along the entire length of the Barrett’s segment) [29].

## 2. Studies Focusing on Diagnostics of BE and EAC Using CLE

The first study with eCLE was published by Kiesslich et al. in 2006. Sixty-three patients with BE were included. The results of this study showed that BE and Barrett’s-associated neoplastic changes could be diagnosed with high accuracy (96.8% and 97.4%) [30].

In 2009, Bajbouj et al. [31] presented a study comparing pCLE with the standard four-quadrant biopsy according to the Seattle Protocol. The study enrolled 68 patients with known or suspected BE. Assessment of pCLE was done first on-site and later, a second time, blind. pCLE recordings were interpreted live during endoscopic examination as well as blind at least 3 months after the endoscopy (the blind analysis was done without the knowledge of any endoscopic or histologic data). Specificity and negative predictive value of pCLE in excluding neoplasia were 97% and 93% for blinded evaluation and 95%/92% for the on-site assessment, but positive predictive values and sensitivity were relatively low (28%/46% for blinded and 12%/18% for the on-site assessment). Although the authors mentioned that pCLE seems to have the potential to improve imaging techniques for surveillance of BE, it was their opinion that pCLE was in need of further improvement and technical development. On the other hand, the results of a study published by Wallace et al. [32] in 2010 showed pCLE to have very high accuracy (92%) for the diagnosis of neoplasia in BE. Endomicroscopists with prior pCLE experience had an overall sensitivity of 91% and specificity of 100%.

Sharma et al. [33] published an international multicentric randomized prospective study in 2011 covering 101 patients with BE, all of whom underwent examination by HD-WLE, narrow-band imaging (NBI), and pCLE. The results were then compared. Sensitivity for HD-WLE was 34.2% and its specificity was 92.7%. For HD-WLE or pCLE, sensitivity was 68.3% and specificity 87.8%. The sensitivity for HD-WLE or NBI was 45.0% and the specificity 88.2%. By comparison, for HD-WLE or NBI or pCLE, sensitivity was 75.8% and specificity 84.2%. These results show that the use of pCLE combined with HD-WLE significantly improved the ability to detect neoplasia in BE patients.

Dolak et al. [34] presented a prospective pilot study in 2014 involving 38 patients with BE-associated neoplasia. All were planned to undergo endoscopic mucosal resection or endoscopic submucosal dissection. First, they underwent HD-WLE with NBI. Then, eCLE mapping of suspected neoplastic lesions was performed by another endoscopist. In 7 of 38 patients (18%), eCLE revealed additional neoplastic tissue compared with prior white-light endoscopy and NBI—: 2 concomitant lesions, 2 cases of lateral tumor extension within the Barrett’s epithelium, and 3 cases of previously undetected subsquamous tumor extension. In conclusion, eCLE was a supporting diagnostic method for planning endoscopic resection by assessing lateral and subsquamous tumor extension of BE-associated neoplasia. This was only a pilot study, however, and therefore the results should be interpretated cautiously.

In 2012, Jayasekera et al. [35] published a cross-sectional study wherein 50 patients with BE were included. All underwent HD-WLE, followed by NBI, and finally eCLE. For the detection of HGD/intramucosal cancer, the sensitivity, specificity, and accuracy were: HD-WLE, 79.1%, 83.1%, and 82.8%; NBI, 89.0%, 80.1%, and 81.4%; and CLE, 75.7%, 80.0%, and 79.9%, respectively. This study concluded that the most accurate method in detecting HGD is HD-WLE in combination with NBI (followed by targeted biopsies).

Bertani et al. [36] published a study in 2013 assessing 100 patients with BE and comparing the incident dysplasia detection rate of biopsies obtained by HD-WLE or by pCLE. Patients were divided into two groups: 50 underwent HD-WLE only and 50 underwent pCLE in addition to HD-WLE. The dysplasia detection rate was significantly higher in the pCLE group than in the HD-WLE group (*p* = 0.04).

Additional studies of CLE in the endoscopic management of Barrett’s dysplasia were published by Caillol et al. [37] in 2017 and Canto et al. [38] in 2014. In the first of these, a retrospective study [37], there were 31 patients and data were collected from 35 endoscopic examinations from 2013 to 2015. The histological results from the endoscopic resections were normal/inflammatory in 3 cases, nondysplastic BE with intestinal metaplasia in 8 cases, LGD in 10 cases, and HGD/EAC in 14 cases. Correct diagnoses were made in 71% (25/35) of the cases by pCLE and in 43% (15/35) of the cases by pre-resection biopsy. The sensitivity, specificity, and accuracy for the detection of HGD/EAC were 92.9%, 71.4%, and 80% for pCLE and 78.6%, 61.9%, and 68.6% for histological biopsy, respectively. However, the differences in favor of pCLE were not statistically significant (*p* = 0.25). The second study by Canto et al. [38] included 192 patients. Patients were randomized to HD-WLE + RB (random biopsy) (Group 1) or HD-WLE + eCLE + TB (targeted biopsy) (Group 2). The addition of eCLE to HD-WLE increased the sensitivity for neoplasia detection to 96% from 40% (*p* < 0.0001). Moreover, CLE changed the treatment plan in 36% of patients.

In 2017, Shah et al. [39] published a study that compared pCLE with random biopsies. The study included 66 patients. Patients underwent HD-WLE and NBI followed by pCLE. Of the 66 patients, 4.55% had HGD or cancer. Both real-time and blinded pCLE correctly identified all cases of cancer. In summary, this study proved that pCLE demonstrates high specificity (98%) for detecting dysplasia and cancer, but lower sensitivity (67%) may limit its utility in routine BE surveillance.

In 2018, Richardson et al. [40] presented a study examining the role of pCLE in BE screening and surveillance as compared to the Seattle Protocol. The 172 patients in that study underwent biopsy according to the Seattle Protocol and pCLE. Tissue biopsy using the Seattle Protocol identified intestinal metaplasia (IM) in 46/172 patients, while pCLE identified intestinal metaplasia (IM) in 99/172 patients (*p* < 0.0001). This study showed pCLE to be considerably more sensitive in detecting BE than is the Seattle Protocol, because many patients with BE were diagnosed positive for IM by pCLE despite negative histology.

The first experiences with pCLE in BE and EAC diagnostics within the Czech Republic were published by Kunovsky et al. in 2020 [41]. This pilot prospective study from January to July 2019 had enrolled 14 patients. Of these, 3 had reflux esophagitis, 4 BE, 3 EAC, and 4 were healthy cohorts. The correct diagnoses based on real-time pCLE were evaluated by an endoscopist in 11 of 14 cases (78.6%).

In 2020, Kollar et al. [42] presented a prospective study with 67 patients with esophageal and/or gastric lesions. The patients underwent high-resolution endoscopy, and lesions were examined by pCLE followed by standard biopsies. pCLE diagnosis was correct in 89.2% of cases; diagnosis based on biopsy was correct in 85% (*p* = 0.6). In conclusion, this study showed that pCLE provides satisfactory diagnostic accuracy comparable with that of standard biopsies in patients with esophageal or gastric lesions.

In 2015, di Pietro et al. [43] presented a study that had included 55 patients. All of them underwent HD-WLE followed by autofluorescence imaging (AFI). After that, AFI-targeted areas were examined by NBI with magnification and also by pCLE. Finally, there were testing biopsies with a molecular panel comprising aneuploidy plus cyclin A and p53 immunohistochemistry. In the per-patient analysis, the overall sensitivity and specificity of AFI-targeted pCLE were 100% and 53.6% for HGD/intramucosal cancer and 96.4% and 74.1% for any grade of dysplasia, respectively. The addition of a three-biomarker panel further improves the diagnostic accuracy for any grade of dysplasia.

A similar study was published in 2022 by Vithayathil et al. [44], where they compared the diagnostic accuracy of AFI-guided pCLE and molecular biomarkers (p53 and cyclin A by immunohistochemistry, aneuploidy by image cytometry) vs. HD-WLE with Seattle protocol biopsies in patients with BE. A total of 134 patients underwent both of these examinations. A finding of this study was that AFI-guided pCLE has similar diagnostic accuracy for dysplasia as does HD-WLE with Seattle protocol biopsies (74.3%; 95% CI: 56.7–87.5 vs. 80.0%; 95% CI: 63.1–91.6; *p* = 0.48). Nevertheless, the addition of molecular biomarkers can improve the diagnostic accuracy.

A summary of studies using CLE in BE and EAC diagnostics is presented in Table 2.

## 3. Meta-Analysis and Systematic Reviews Focusing on Diagnostics of BE and EAC Using CLE

In 2014, two meta-analyses were published by Wu et al. [49] and Gupta et al. [50]. The first-mentioned of these [49] included 8 studies with 709 patients and 4008 specimens. The pooled sensitivity of CLE for detection of neoplasia was 89% (per-patient analysis) and 70% (per-location analysis). In the second meta-analysis by Gupta et al. [50], there were 7 studies with 345 patients and 3080 lesions. The authors compared the diagnostic accuracy of CLE with that of targeted biopsies and with standard four-quadrant random biopsies in the detection of HGD/EAC. In conclusion, CLE seems to be a valid diagnostic method, especially in identifying HGD/EAC. Because of its relatively low sensitivity of 68% (95% CI of 64–73%) and positive likelihood ratio of 6.56 (95% CI of 3.61–11.90), however, CLE may not replace standard biopsy techniques at present and more study is needed to prove if it is a reliable method.

By contrast, Xiong et al. [45] published a meta-analysis about CLE in 2015 that covered 14 studies and included 789 patients with 4047 lesions. Two commercially available CLE systems (eCLE and pCLE) were used. This study concluded that CLE seems to be a suitable method for differentiating neoplasms from non-neoplasms in BE. In those authors’ opinions, CLE could be used for BE surveillance and early diagnosis of EAC because of its high sensitivity of 89% (95% CI: 0.82–0.94, I^2^ = 31.6%) and accuracy.

In 2018, Xiong et al. [51] published another meta-analysis where NBI and CLE were used for the detection of neoplasia in BE. This meta-analysis covered 5 studies involving 251 patients. The meta-analysis showed that CLE compared to NBI significantly increased the per-lesion detection rate of BE-associated esophageal neoplasia, HGD, and EAC. Compared with NBI, the pooled additional detection rate of CLE for per-lesion detection of neoplasia in patients with BE was 19.3% (95% CI: 0.05–0.33, I^2^ = 74.6%). However, the pooled sensitivity and specificity of CLE were similar to those for NBI at the per-lesion level (72.3% vs. 62.8% and 83.8% vs. 85.3%, respectively).

The American Society for Gastrointestinal Endoscopy (ASGE) Technology Committee regularly performs systematic reviews and meta-analyses to evaluate endoscopic technologies for determining whether these have met previously established Preservation and Incorporation of Valuable endoscopic Innovations (PIVI) thresholds. In 2016, the ASGE Technology Committee published a systematic review and meta-analysis [52] confirming that the thresholds set by ASGE PIVI for real-time imaging-assisted endoscopic targeted biopsy during endoscopic surveillance of BE have been met by acetic acid chromoendoscopy, NBI, and eCLE. The pooled sensitivity for eCLE was 90.4% (95% CI: 72–97, I^2^ = 79) and the pooled specificity for eCLE was 92.7% (95% CI: 87–96, I^2^ = 0). Results with autofluorescence imaging and pCLE are encouraging, although these do not yet meet the established PIVI thresholds. The pooled sensitivity for pCLE was 90.3% (95% CI: 72–99, I^2^ = 93) and the pooled specificity for pCLE was 77.3% (95% CI: 54–91, I^2^ = 88).

## 4. Surveillance of BE by CLE and Surveillance by CLE of BE Neoplastic Lesions after Endoscopic Treatment

In almost all studies and meta-analyses the role of CLE in BE surveillance was discussed. Due to a lack of scientific evidence, however, the standard method today for BE surveillance is HD-WLE with biopsies. Surveillance intervals are variable for different BE lengths according to the ESGE recommendations: For patients with irregular Z-linie/columnar lined esophagus < 1 cm, no routine biopsies or endoscopic surveillance is advised. For BE ≥ 1 cm and <3 cm, BE surveillance should be repeated every 5 years. For BE ≥ 3 cm and <10 cm, the interval for endoscopic surveillance should be 3 years. Patients with BE ≥ 10 cm, should be referred to a BE expert center for surveillance endoscopies [53].

Wallace et al. published in 2012 [54] a randomized multicenter study assessing whether use of pCLE in addition to HD-WLE could aid in the determination of residual BE after ablation treatment. A total 119 patients with BE undergoing ablation were enrolled. The overall primary outcome of optimal treatment was achieved for 26% (15/57) of patients in the HD-WLE arm and for 27% (17/62) in the HDWLE + pCLE arm. This study yields no evidence that the addition of pCLE to HDWL imaging for detection of residual BE or neoplasia can provide improved treatment.

In 2020, Krajciova et al. [55] presented a single center prospective study that included 56 patients with diagnoses of LGD, HGD, and EAC and who underwent endoscopic treatment (endoscopic mucosal resection or dissection, radiofrequency ablation). The patients underwent surveillance endoscopy with pCLE followed by standard biopsies. In conclusion, pCLE was at least as effective as standard biopsies in detecting persistent/recurrent intestinal metaplasia after endoscopic treatment of BE neoplasia.

## 5. Advantages and Disadvantages

While the use of high-definition endoscopy is strongly recommended by ESGE, CLE still does not have a place in the surveillance of BE. Comparing these two methods, pCLE’s main advantage is that no tissue sampling is needed, and so there is also lower risk of adverse effects, such as bleeding. The risk of bleeding from standard biopsies is nevertheless very low, and this advantage is more or less restricted to patients on anti-platelet or anti-coagulation therapy. Another advantage is that examination by pCLE enables quicker determination of diagnosis than does HD-WLE [56,57]. On the other hand, pCLE’s greatest disadvantage is its high cost, which can be a significant barrier to widespread clinical implementation. Other disadvantages can relate to availability, physician training in image interpretation, the extra time needed to view the images during endoscopy, and the role of the pathologist [58,59,60]. Using pCLE also could have a side effect in cases of allergy to the contrast agent, fluorescein, which is used to stain the tissues [61,62]. However, the safety of fluorescein for pCLE was demonstrated in a large study by Wallace et al. in 2010 that included about 2300 gastrointestinal endomicroscopy procedures. Mild side effects, such as hypotension, nausea, rash, and mild epigastric pain occurred in only 1.4% of patients [63]. Some studies on colon neoplastic lesions and precancerous or early stage esophageal squamous cancer that pCLE can be learned quickly in a short period of time [64,65].

## 6. Conclusions

Numerous studies have proven pCLE to be a diagnostic method with high accuracy for the detection of neoplastic lesions in the esophagus and malignant transformations of BE. Moreover, a big advantage of this technique is that it is noninvasive, not necessitating the taking of biopsies. Nevertheless, pCLE is still not used as a diagnostic method in the surveillance of BE and it cannot yet replace standard histopathology. More prospective randomized control trials and high-quality meta-analysis are needed to provide scientific evidence that can be used to integrate pCLE as a diagnostic method into the surveillance of BE.

## Figures and Tables

**Figure 1 diagnostics-12-01616-f001:**
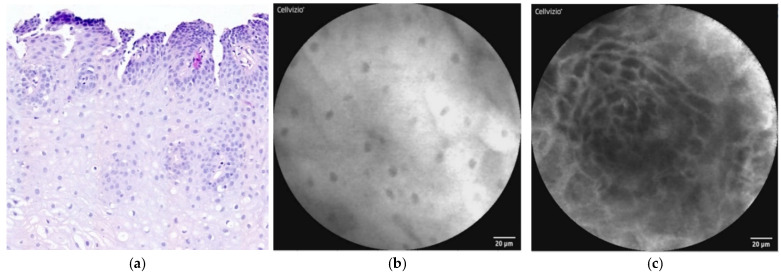
(**a**) Histopathology: image of stratified, nonkeratinizing squamous cell epithelium of distal esophagus with stromal papillae, hematoxylin–eosin staining. (**b**,**c**) pCLE view: squamous cells of esophageal epithelium (**b**) with stromal papillae (**c**).

**Figure 2 diagnostics-12-01616-f002:**
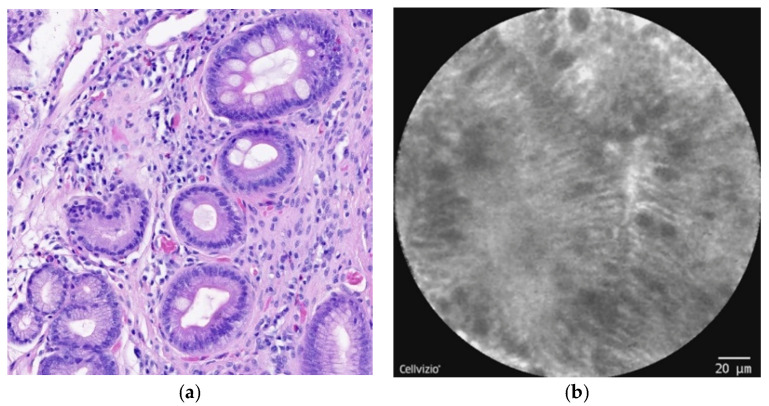
(**a**) Histopathology: nondysplastic BE with intestinal metaplasia of the esophageal epithelium, cylindrical unciliated epithelium with goblet cells. Without dysplastic changes. Hematoxylin–eosin staining. (**b**) pCLE view: intestinal metaplasia of the esophageal epithelium, columnar cells with dark goblet cells.

**Figure 3 diagnostics-12-01616-f003:**
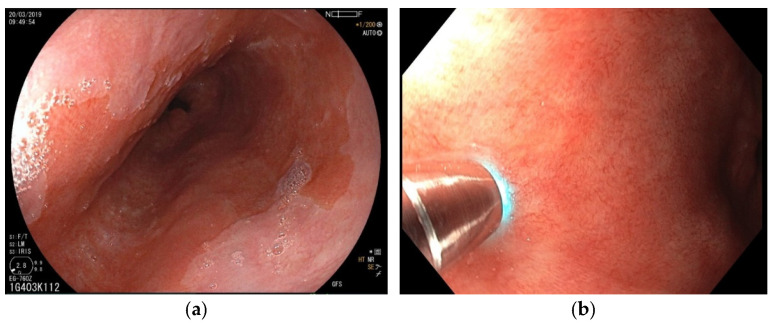
(**a**) Endoscopic view of the long segment of BE. (**b**) Endoscopic view of nondysplastic BE investigated with pCLE. The probe attached to the mucosa can be seen.

**Figure 4 diagnostics-12-01616-f004:**
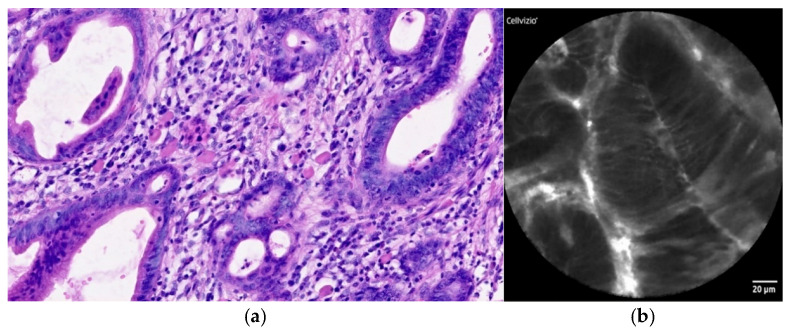
(**a**) Histopathology: EAC consisting of tubular and cribriform glandular formations with intestinal-type epithelium with marked cytologic atypia. The stroma is desmoplastic. Hematoxylin–eosin staining. (**b**) pCLE view: irregular glandular structures, lined by atypical cylindrical cells. Loss of goblet cells is also present.

**Table 1 diagnostics-12-01616-t001:** CLE diagnostic criteria for nondysplastic BE and dysplastic BE/EAC (table created according to Wallace et al. [10], di Pietro et al. [12], and Gaddam et al. [13]).

Normal Squamous Epithelium	Nondysplastic BE	LGD *	HGD/EAC **
Flat cells without crypts or villi	Uniform villiform architecture	Dark non-round glands	Saw-toothed epithelial surface
Bright vessels within papillae (intrapapillary capillary loops)	Columnar cells	Irregular gland shape	Unequal size and shape of glands
	Dark goblet cells	Lack of goblet cells	Not-easily identifiable goblet cells
		Sharp cutoff of darkness	Non-equidistant glands
		Variable cell size	Enlarged cells
		Cellular stratification	Irregular and non-equidistant cells

* Cutoff for LGD diagnosis is positivity of any three of the six criteria. ** Two or more of these pCLE criteria need to be met for the diagnosis of HGD in BE patients. Using these validated criteria, the overall accuracy for diagnosis of HGD/cancer was >80%.

**Table 2 diagnostics-12-01616-t002:** Summary of CLE studies in BE and EAC diagnostics (table created according to Vranic et al. [28] and Xiong et al. [45]).

Authors	Year	Type of CLE	No. of Patients	Sensitivity (%) *	Specificity (%) *
Kiesslich et al. [30]	2006	eCLE	63	93	98
Pohl et al. [46]	2008	pCLE	75	75	58
Bajbouj et al. [31]	2009	pCLE	68	90	59
Wallace et al. [32]	2010	pCLE	5	88	96
Sharma et al. [33]	2011	pCLE	101	100	56
Gaddam et al. [13]	2011	pCLE	122	76	85
Jayasekera et al. [35]	2012	eCLE	50	76	80
Trovaro et al. [47]	2013	eCLE	48	83	95
Bertani et al. [36]	2013	pCLE	100	100	83
Canto et al. [38]	2014	eCLE	192	100	95
Dolak et al. [34]	2014	eCLE	38	-	-
Legget et al. [48]	2016	pCLE	27	76	79
Caillol et al. [37]	2017	pCLE	31	93	71
Shah et al. [39]	2017	pCLE	66	67	98
Richardson et al. [40]	2018	pCLE	172	-	-
Kunovsky et al. [41]	2020	pCLE	14	-	-
Kollar et al. [42]	2020	pCLE	67	88	92

* Sensitivity and specificity for the detection of BE and EAC.

## Data Availability

Not applicable.

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
