# Peer review of "The Use of Confocal Laser Endomicroscopy in Diagnosing Barrett’s Esophagus and Esophageal Adenocarcinoma"

_diagnostics, 2022, doi:10.3390/diagnostics12071616_

Round 1
Reviewer 1 Report
This review provides a summary of the evidence on the use of CLE for Barrett's neoplasia.
This review feels like a catalogue of result from previous studies and has very little of the authors opinion on the evidence or attempt to help the reader to take the key elements from individual publications. Results from large RCTs and very small cohort studies are presented with very little emphasis on the vastly difference quality of the evidence published.
There are several important papers which are not described in the text including a recent RCT and the paper on the diagnostic criteria
Here below my specific comments.
Introduction
Page 2, lines 55-59: provide more details of the intraoperative study. What cancer were included, how was pCLE performed?
Page 2 line 60-61. Specify what disease were covered by the Miami classification consensus
Page 2 64-66: provide reference in support of this statement, I am not familiar with the 30-fold increase in the cancer risk on BE patients
Page 2 line 68: mucous should read “Mucosa”
Page 2 line 79: intestinal metaplasia can be present also in dysplastic BE. This sentence is best removed
Section 2
It would be good to mention in the individual studies the proportion of patients with dysplasia included as most studies are on enriched populations
Page 4 line 113: the authors should specify what blinded and on-site assessment are.
Page 4 line 113-116. It would be appropriate not to report the opinion of the authors from another publication, but rather the authors’ own opinion in this context
Page 4 line 122-126. The authors appear to quote text from the Sharma paper. This is not appropriate.
Page 4 lines 120 -128: the design of the Sharma paper including the type of randomisation should be described in order to understand the results.
Page 4-5 line 134-137. This is only a pilot study, therefore caution is needed in the intrastation of the results
Page 5 line 141: what type of CLE was used in this study?
Table 1: the authors should specify what are sensitivity and specificity for
Page 5 line 180-182. Not sure it is important to specify that the Kunovsky study present the first experience in Czech Republic
I would recommend to split the results in section 2 separating eCLE and pCLE as these are two different techniques
The authors only mention Miami classification. However this is not the most commonly used in Barrett’s neoplasia. There are more recent classification by Gaddam et al for HGD/EAC and di Pietro et al for LGD. This should be referenced and a table with different criteria included would be appropriate. It would be also appropriate to dedicate a section to the diagnostic criteria and provide some illustration of CLE images corresponding to different grades of disease.
The result of a recent RCT by di Vithayathil et al 2022 are missing. There is also a cohort study by di Pietro et al (2015) using pCLE in combination with AFI which is missing.
Section 3
The results on the diagnostic accuracy of the Gupta meta-analysis should be presented
Line 209-211. Again avoid referring to other authors opinion
Section on meta-analysis: it would be good to understand the differences in terms of methodology and aims among the Wu, Gupta and Xiong meta-analyses
Page 7, Line 230-235, please. Specify that the study was terminated pre-maturely due to lack of difference between the two groups
It would be appropriate to refer to the outcome of the ASGE PIVI consenus on CLE
Section 4
Page 7 line 237 – 242. I am not sure in what the study by Kraychova was contrasting as it also showed on difference between pCLE and standard biopsies protocol
Section 5
Change title to advantages and disadvantages
The risk of bleeding from standard biopsies is very small. Could this advantage be restricted to patients on anti-platelet or anti-coagulation?
The study by Vithayathil et al showed there pCLE is more time consuming than WLE, therefore the statement of procedure time is questionable
The authors should comment on the learning curve for CLE and the data available.
Section 6
Given the large body of evidence available already the authors should provide their opinion on what type of study is actually missing in order to support its use in clinical practice.
Author Response
Dear reviewer.
Thank you so much for your careful review and recomendations. The suggested changes were done, please see a comments in attached file - point by point response.
best regards
Lumir Kunovsky

Reviewer 2 Report
Dear Editor,
It was my pleasure to be asked to read and evaluate this work titled "The use of confocal laser endomicroscopy in diagnosing Barrett’s esophagus and esophageal adenocarcinoma". Vaculová et al. presented a good review of the use of confocal laser endomicroscopy in Barrett's esophagus and esophageal adenocarcinoma.
I think that the presented nice study will contribute to the literature. Congratulations to the authors for the good work.
Sincerely
Author Response
Dear Reviewer,
thank you very much for your positive review.
best regards
Lumir Kunovsky
Reviewer 3 Report
It is an interesting Review on the role of of pCLE in the diagnosis of Barrett's Esophagus and Esophageal Adenocarcinoma.
My suggestions are the following:
- the sentence -“The sensitivity and specificity for HD-WLE were 34.2% and 92.7%, respectively, compared with 68.3% and 87.8%, respectively, for HD-WLE or pCLE (p = 0.002 and p < 0.001, respectively). The sensitivity and specificity for HD-WLE or NBI were 45.0% and 88.2%, respectively, compared with 75.8% and 84.2%, respectively, for HD-WLE, NBI, or pCLE (p = 0.01 and p 1= 0.02, respectively) [29].”- does not seem to be correct; probably the "or" needs to be changed in "and"; please check it.
- the final sentence "More study is needed to provide scientific evidence that can be used to integrate pCLE as a diagnostic method into the surveillance of BE." seems to be too general; I think you should suggest wich type of study or scientific evidence should be obtained to better define the role of pCLE in BE surveillance.
- english language and style could be improved
Author Response
Dear reviewer,
thank you so much for your review and comments.
1) I double-checked the article and the sentence is correct as it is written. (Sharma P, Meining AR, Coron E, Lightdale CJ, Wolfsen HC, Bansal A, Bajbouj M, Galmiche JP, Abrams JA, Rastogi A, Gupta N, Michalek JE, Lauwers GY, Wallace MB. Real-time increased detection of neoplastic tissue in Barrett's esophagus with probe-based confocal laser endomicroscopy: final results of an international multicenter, prospective, randomized, controlled trial. Gastrointest Endosc. 2011 Sep;74(3):465-72. doi: 10.1016/j.gie.2011.04.004. Epub 2011 Jul 13.)
2) The type of studies was specified.
3) The English language will be checked once again by a native speaker.
best regards
Lumir Kunovsky
Round 2
Reviewer 1 Report
The manuscript has significantly improved compared to the previous version. Most of the comments have been satisfactorily addressed however I would like to point the author attention on two comments that I cannot see have been appropriately address
Page 5 line 136: the authors should specify what blinded and on-site assessment are. I cannot see this being addressed in the revised version
Page 6 line 147. The authors appear to quote text from the Sharma paper. This is not appropriate. There remains plagiarism in writing with text copied and pasted from the abstract of the original paper. The text should be modified to avoid plagisrism
Author Response
Page 5 line 136: the authors should specify what blinded and on-site assessment are. I cannot see this being addressed in the revised version.
The information has been specified.
Page 6 line 147: The authors appear to quote text from the Sharma paper. This is not appropriate. There remains plagiarism in writing with text copied and pasted from the abstract of the original paper. The text should be modified to avoid plagisrism
This part was rewritten.